# Irisin Enhances Mitochondrial Function in Osteoclast Progenitors during Differentiation

**DOI:** 10.3390/biomedicines11123311

**Published:** 2023-12-14

**Authors:** Eben Estell, Tsunagu Ichikawa, Paige Giffault, Lynda Bonewald, Bruce Spiegelman, Clifford Rosen

**Affiliations:** 1Center for Molecular Medicine, MaineHealth Institute for Research, Scarborough, ME 04074, USAclifford.rosen@mainehealth.org (C.R.); 2Department of Anatomy, Cell Biology and Physiology, Orthopaedic Surgery, School of Medicine, Indiana University, Indianapolis, IN 46202, USA; lbonewal@iu.edu; 3Indiana Center for Musculoskeletal Health, Indiana University, Indianapolis, IN 46202, USA; 4Department of Cancer Biology, Dana-Farber Cancer Institute, Harvard Medical School, Boston, MA 02115, USA; bruce_spiegelman@dfci.harvard.edu; 5Department of Cell Biology, Harvard Medical School, Boston, MA 02115, USA

**Keywords:** irisin, osteoclast, osteoclast progenitor, mitochondria, oxidative respiration

## Abstract

Irisin is a myokine released from muscle during exercise with distinct signaling effects on tissues throughout the body, including an influence on skeletal remodeling. Our previous work has shown that irisin stimulates resorption, a key first step in bone remodeling, by enhancing osteoclastogenesis. The present study further investigates the action of irisin on the metabolic function of osteoclast progenitors during differentiation. Fluorescent imaging showed increased mitochondrial content and reactive oxygen species production with irisin treatment in osteoclast progenitors after 48 h of osteoclastogenic culture. Mitochondrial stress testing demonstrated a significant increase in maximal oxygen consumption rate and spare capacity after 48 h of preconditioning with irisin treatment. Together, these findings further elucidate the stimulatory action of irisin on osteoclastogenesis, demonstrating an enhancement of metabolism through mitochondrial respiration in the progenitor to support the energy demands of their differentiation into mature osteoclasts.

## 1. Introduction

Irisin is a myokine, or muscle-derived signaling factor, first described to release from muscle during exercise and stimulate adipose browning and thermogenesis [1]. A growing body of research has demonstrated potent cellular effects of irisin on cells across a range of tissues, including a role in bone homeostasis and response to exercise and loading. Early studies demonstrated an anabolic effect of irisin in bone, with elevated levels of irisin in serum from exercised mice stimulating osteoblast differentiation in vitro [2]. In vivo models then demonstrated that intermittent injections of irisin both increased cortical bone mass [3] and protected against unloading-induced bone loss [4]. Our own work has since suggested that irisin influences both sides of the remodeling equation, stimulating osteoclastogenesis and resorption as well [5]. These initial studies characterized an increase in mature osteoclasts after 7 days of growth factor-induced differentiation from primary bone marrow progenitors. An upregulation of early osteoclast markers as well as the ability of brief irisin exposures to yield higher osteoclast numbers a week later both pointed to a rapid action of irisin on the osteoclast progenitor with a lasting stimulatory effect on differentiation. To better understand the mechanisms by which irisin promotes osteoclast differentiation, this study investigates the effect of irisin on the metabolic function of osteoclast progenitors during the early stages of osteoclastogenesis.

Osteoclasts have long been characterized as highly energy-demanding cells, both during their differentiation from macrophage-lineage bone marrow progenitors and in their key function of bone resorption as mature cells. These energy demands are supported by enhanced glucose metabolism, with early work demonstrating that exogenous ATP promotes both osteoclast differentiation and resorption [6]. This enhanced glucose metabolism relies on both mitochondrial respiration and glycolysis at different stages of osteoclast maturation. Early work by Baron et al. described the osteoclast progenitor as a mitochondria-rich cell with high levels of enzymes used in oxidative phosphorylation (OxPhos), a key process of mitochondrial respiration [7]. Further work has supported the tenet that osteoclast progenitors rely on mitochondrial respiration, showing a substantial increase in mitochondria number during differentiation [8], and that promotion of OxPhos with pyruvate enhances osteoclastogenesis while pharmacological blocking of mitochondrial ATP production inhibits this differentiation [9].

The accumulation of reactive oxygen species (ROS), a product of mitochondrial OxPhos, has also been implicated as a key step during osteoclastogenesis. RANKL signaling has been specifically shown to increase ROS levels in differentiating bone marrow progenitors and RAW264.7 monocytes, and blocking ROS production inhibited RANKL-induced osteoclastogenesis, indicating that ROS play a key role as an intercellular signaling mediator in differentiating osteoclasts [10]. Other work has shown similar increases in mitochondrial ROS during osteoclast differentiation [11,12].

Taken together, these findings demonstrate the importance of mitochondrial respiration in the differentiative processes that support osteoclast formation, bone resorption, and ultimately coupled remodeling. The present study seeks to further clarify the impact of irisin in this context with the hypothesis that irisin enhances progenitor mitochondrial function to support the energy demands of differentiation and thereby enhance the formation of mature osteoclasts.

## 2. Materials and Methods

Osteoclast Progenitor Culture: In vitro, differentiation of primary osteoclast progenitors was performed via established protocols from bone marrow-derived hematopoietic stem cells [5]. Briefly, bone marrow was isolated via centrifugation from the femur and tibia of 8-week-old male C57BL/6J mice (Jackson Laboratories, Bar Harbor, ME, USA), with the non-adherent cell fraction cultured in MEM-α (10% FBS/1% Pen-Strep) supplemented with 100 ng/mL of RANKL and 30 ng/mL of M-CSF (PeproTech, Cranbury, NJ, USA to induce osteoclastogenesis. Primary cells were plated at 5 × 10^4^ cells/well in tissue-culture-treated 96-well plates for fluorescence imaging or in Seahorse XFe 96-well microplates for metabolic analysis. Differentiative cultures were maintained for 48 h to study the effect of irisin on the progenitor population prior to the formation of mature osteoclasts. Recombinant irisin, obtained as previously described [13], was supplemented at 10 ng/mL in the media continuously for 48 h and removed during subsequent experimentation to investigate the persistent effect of preconditioning compared to untreated controls. All primary cell experiments were conducted in duplicate (Appendix A), with bone marrow progenitors derived from a pooled cell population from three littermate mice, utilizing two separate litters at different times.

To address the heterogenous nature of primary cell isolations and further confirm the reproducibility of results, the RAW 264.7 macrophage-like cell line (ATCC) was utilized for a set of additional experimental repeats (Appendix A). In vitro, osteoclast differentiation of RAW cells was performed by previously established protocols [14,15]. Briefly, RAW cells were maintained and passaged in high-glucose (4.5 g/L) DMEM containing 10% fetal bovine serum (FBS) and 1% Pen-Strep and were plated for experiments in MEM-α (10% FBS/1% Pen-Strep) supplemented with 100 ng/mL of RANKL to induce osteoclastogenesis. RAW cells were plated at 8 × 10^3^ cells/well in tissue-culture-treated 96-well plates for fluorescence imaging and 2 × 10^3^ cells/well in Seahorse XFe 96-well microplates for metabolic analysis to accommodate high levels of basal respiration.

Cellular Fluorescence Imaging: Cellular mitochondria content and reactive oxygen species (ROS) concentration were assessed via fluorescence imaging with 200 nM of MitoTracker Red CMXRos (Invitrogen, Carlsbad, CA, USA) and 5 µM of CellROX (Invitrogen), respectively. Dyes were loaded for 30 min at 37 °C before imaging at 10× magnification on a fluorescence microscope (EVOS M5000 Imaging System, Invitrogen) for cellular fluorescence quantification. Images were acquired for a fixed central region in each of the 8 replicate wells per group, with acquisition settings fixed for all samples across each dye. Average cellular fluorescence intensity for individual cells (500–2000 per group, pooled across replicate wells) was analyzed in ImageJ, normalized to the average of six background regions for each image, and calculated as corrected total cell fluorescence (CTCF), where CTCF = integrated density—(cell area × mean background fluorescence) [16]. Additional images of individual cells were captured at 63× on a confocal microscope (Leica, Wetzlar, Germany) to qualitatively assess the density of mitochondria.

Real-Time Metabolic Analysis: Utilizing a Seahorse XF96 (Agilent Technologies, Santa Clara, CA, USA), mitochondrial stress testing was performed via pharmaceutical decoupling of the electron transport chain with the sequential addition of 1 µM of Oligomycin (OLIGO), 2 µM of Carbonyl cyanide-4 (trifluoromethoxy) phenylhydrazone (FCCP), and 1 µM of Rotenone/Antimycin (ROT/AA). Oxygen consumption rates (OCR) were measured three times at 6 min intervals for an initial baseline period and then after the injection of each drug. Results were normalized to cell number via visualization of nuclei via Hoechst Dye and automated counting with a Cytation 1 Imaging Reader (BioTek, Winooski, VT, USA). Key respiratory parameters were calculated from real-time OCR data as follows:Non-mitochondrial OCR (Non-Mito): minimum rate after ROT/AA;Basal Respiration: last rate before Oligo—Non-Mito;Maximal Respiration: maximum rate after FCCP—Non-Mito;Proton Leak: minimum rate after Oligo—Non-Mito;ATP-linked Respiration: last rate before Oligo—minimum rate after Oligo;Spare Respiratory Capacity: maximal—basal.

Seahorse microplates were organized such that 23 wells per group were arranged symmetrically (each taking up a quadrant) to avoid any variation in wells on the edge of the plate. Individual wells were included in the post-analysis based on the criteria that they had repeatable rate measurements at baseline and after each drug injection and that they responded to each drug.

Statistical Analysis: Statistical comparison between control and 48 h irisin treatment was conducted in Prism 10.1.1 (GraphPad, La Jolla, CA, USA) via unpaired two-tail Student’s *t*-test for fluorescence imaging and two-way ANOVA with Sidak correction for multiple comparisons for respiration parameters. For both analyses, a *p*-value of less than 0.05 was required for statistical significance.

## 3. Results

### 3.1. Irisin Increases Osteoclast Progenitor Mitochondria and Reactive Oxygen Species Content

Mitochondrial content was assessed in osteoclast progenitors using the MitoTracker Red CMXRos fluorescent dye, which accumulates in live cell mitochondria in a membrane potential-dependent manner. Fluorescence imaging demonstrated a qualitative increase in density and fluorescence of mitochondria within osteoclast progenitors treated with irisin for 48 h compared to untreated controls (Figure 1a–d). This increase in mitochondrial number and membrane potential (and therefore fluorescence) was quantified as a significant increase in the mean corrected total cell fluorescence of individual cells (*p* < 0.0001, Figure 1e).

Mitochondrial function was further assessed with respect to the cellular production of reactive oxygen species (ROS) using the CellROX Green dye. Representative images show a qualitative increase in fluorescence with 48 h irisin treatment (Figure 2a,b). Quantitative analysis of mean corrected total cell fluorescence demonstrates a significant increase with irisin treatment (*p* < 0.0001, Figure 2c).

### 3.2. Irisin Enhances Osteoclast Progenitor Mitochondrial Respiration

Mitochondrial stress testing of osteoclast progenitors after 48 h of in vitro differentiation demonstrated a typical metabolic profile with sequential pharmacological inhibition of the electron transport chain (ETC). Untreated progenitors had a steady baseline oxygen consumption rate (OCR) that decreased with ATP synthase inhibition via Oligomycin (OLIGO) to reveal the fraction of OCR contributing to cellular ATP production. A high maximal respiration rate over baseline was then achieved under conditions of uninhibited electron flow through the ETC via disruption of mitochondrial membrane potential with FCCP. A complete shutdown of mitochondrial respiration via rotenone (ROT) and antimycin a (AA) then revealed the underlying portion of OCR contributed via non-mitochondrial processes (Figure 3a). Throughout these stages of mitochondrial stress testing, both irisin and preconditioning increased OCR levels above those observed for untreated controls, particularly evident during maximal respiration (Figure 3a). Considering key respiratory parameters calculated with respect to basal OCR (color panel backdrop, Figure 3a), irisin treatment significantly increased maximal respiration and spare respiratory capacity (*p* < 0.0001, Figure 3b).

## 4. Discussion

The osteoclast progenitor has been described as a mitochondria-rich cell, with a further increase in mitochondrial content over the course of differentiation to support the energy demands of fusion into mature osteoclasts [7,8]. Utilizing a potentiometric mitochondrial membrane dye, we similarly observed that osteoclast progenitors in differentiative conditions (with RANKL and MCSF) for 48 h were qualitatively rich with mitochondria. Treatment with 10 ng/mL of irisin further increased mitochondrial content compared to the untreated controls, as quantified by the fluorescence intensity of individual cells (Figure 1). While the potentiometric nature of the MitoTracker dye offers some insight into the membrane potential of the mitochondria, we sought to gain further insight into the relative productivity of these organelles by measuring reactive oxygen species (ROS), a key output of mitochondrial respiration through the electron transport chain [17]. While often associated with cellular stress when accumulated in greater concentrations, changes in ROS levels have been shown to play a key messaging role during osteoclastogenesis [10], making them a particularly relevant metric for irisin-stimulated mitochondrial respiration in osteoclast progenitors. Following mitochondrial content, we observed that differentiating osteoclast progenitors contained detectable levels of ROS as visualized with a fluorescent indicator and that irisin significantly increased ROS fluorescence intensity in individual cells (Figure 2).

To further assess the function of mitochondria in osteoclast progenitors, we employed real-time monitoring of oxygen consumption rates (OCR) during pharmaceutical uncoupling of the mitochondrial membrane electron transport chain. This mitochondrial stress testing revealed a characteristic energy state for osteoclast progenitors that featured a high maximal respiration rate compared to basal respiration, corresponding to high spare capacity. These results agree with a consensus in the literature that mitochondrial oxidative phosphorylation is an important energy source for osteoclast progenitors during differentiation. Treatment with 10 ng/mL of irisin for the duration of the 48 h differentiative culture enhanced these characteristics with a significant increase in both maximal respiration and spare respiratory capacity (Figure 3). As maximal respiration and spare capacity are both indicators of the energy flexibility of the cell, enhancement of these characteristics by irisin may allow the osteoclast progenitor to better function under stress and meet the energy demands of fusion into mature osteoclasts.

As quantified by automated counting of Hoechst-stained nuclei, the number of progenitors in each well increased above the initial plating density in all groups during 48 h of culture, but irisin treatment had no effect on this proliferation. Taken together with the fact that the metabolic analyses were normalized by cell numbers, this suggests that the larger numbers of mature osteoclasts we previously observed [5] were the result of irisin enhancing the energetic and fusile capacity of available progenitors as opposed to promoting the proliferation of more progenitors. This mechanism of action may be particularly important for the rapid effect of irisin on bone resorption in vivo, whereby irisin released transiently during exercise can have a more immediate stimulatory effect on existing osteoclast progenitors in the marrow and at the bone surface, compared to enhancing the recruitment and proliferation of more progenitors later.

While the present work provides further evidence that irisin targets the osteoclast progenitor, the direct signaling effects of irisin on the mature osteoclast remain to be further clarified. The work of Baron et al. highlights that mature osteoclasts retain a mitochondria-rich characteristic, inheriting the cellular contents of their progenitors [7]. While some work has shown that mature osteoclasts utilize OxPhos to generate the ATP required for resorptive activities [18], others describe a more complete reliance on glycolysis for resorption, demonstrating that osteoclasts forced to rely on OxPhos by inhibiting glycolysis have significantly impaired resorption [19]. Further work has shown increases in extracellular acidification rate (ECAR) in mature osteoclasts, indicative of increased glycolysis [20]. Recent studies of RANK-L and polyethylene-induced osteoclastogenesis in RAW264.7 monocytes have shown increases in both mitochondrial OCR, ROS production, and glycolytic ECAR during osteoclast differentiation [15]. Taken together, this growing body of work highlights a ramping up of cellular glucose metabolism from both mitochondrial and glycolytic processes to support the energy demands of differentiating progenitors and bone-resorbing mature osteoclasts. Future work will leverage the approaches developed in this study to investigate the effect of irisin on both mitochondrial and glycolytic metabolism and the balance between these two processes during differentiation from progenitor to mature osteoclast.

The present work further elucidates the mechanisms by which irisin stimulates osteoclastogenesis, thereby regulating the resorptive processes that are a key first step in bone remodeling. Taken together with previous studies demonstrating an anabolic effect of irisin through the osteoblast [2] and direct signaling effects on the osteocyte, which regulates both osteoclast and osteoblast function [13], it is clear that irisin plays a multi-faceted role within the remodeling unit. Unraveling the net effect of these interactions on bone in a physiologic context has often been challenged by seemingly contradictory results from separate experiments and will require a better understanding of how the dynamics of irisin signaling influence its cellular effects. With respect to the osteoclast alone, while we have shown that continuous physiologic irisin doses increase osteoclast number in vitro, others have shown higher concentrations to be inhibitory [21]. While global deletion of the irisin precursor *Fndc5* was shown to be protective against resorptive bone loss following ovariectomy [13], intermittent irisin injections of 100 µg/kg have also been shown to protect against ovariectomy-induced bone loss [22] and promote bone formation [3], both while reducing osteoclast numbers. The same intermittent injections of irisin were also shown to accelerate bone fracture healing in part by increasing osteoclast numbers in the wound callus at early stages [23], indicating that irisin may have different effects on osteoclasts not only depending on timing and dosage but also on tissue and pathological context.

## 5. Conclusions

In this study we have demonstrated that irisin increases the number and respiratory function of osteoclast progenitor mitochondria. This enhancement of bioenergetic pathways that are key to osteoclast differentiation is one mechanism by which irisin may initiate and influence bone resorption. The full picture of irisin’s physiologic role on bone remodeling however remains to be resolved. We anticipate that a more complete understanding of irisin’s specific cellular signaling effects will help elucidate its overall impact on the complex process of bone remodeling and thus inform future efforts to leverage this peptide as a potential therapeutic to protect bone integrity. While the presence of multiple cell targets within bone complicates the understanding of irisin’s physiologic function, it also presents a potential advantage over current anti-resorptive or anabolic drugs, whose influence on one side of balanced remodeling is a limitation that must be considered during extended use.

## Figures and Tables

**Figure 1 biomedicines-11-03311-f001:**
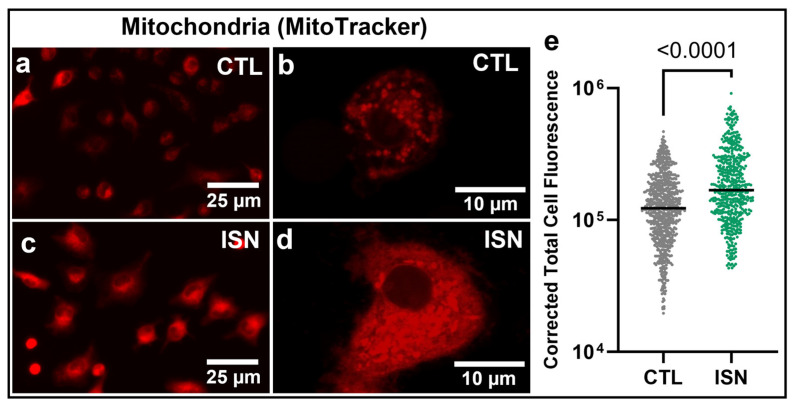
Representative 10× images of fluorescence-labeled mitochondria in untreated (CTL) and 10 ng/mL of irisin-treated (ISN) primary osteoclast progenitors after 48 h of differentiative culture (**a**,**c**). High-magnification 63× imaging shows individual mitochondrial structures (**b**,**d**). Quantification of corrected total cell fluorescence for individual cells pooled across 8 replicate wells (**e**). N = 734 and 509 cells/group for CTL and ISN, respectively. Representative images and quantification are shown for one of three experimental repeats. *p*-value via an unpaired, two-tailed Student’s *t*-test, with significant values (*p* < 0.05) is displayed on the graph.

**Figure 2 biomedicines-11-03311-f002:**
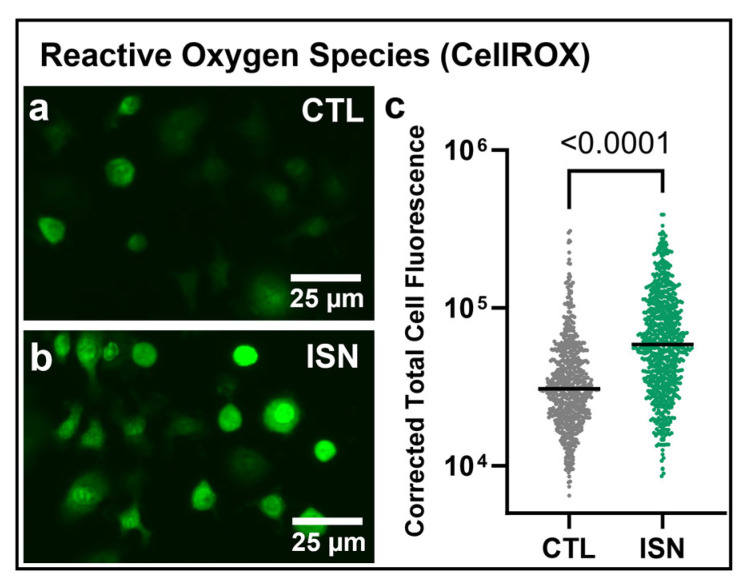
Representative 10× fluorescence imaging of reactive oxygen species concentration in untreated (CTL) or 10 ng/mL of irisin-treated (ISN) osteoclast progenitors after 48 h of culture (**a**,**b**). Quantification of corrected total cell fluorescence for individual cells pooled across 8 replicate wells (**c**). N = 657 and 758 cells/group pooled across wells for CTL and ISN, respectively. Representative images and quantification are shown for one of three experimental repeats. *p*-value via an unpaired, two-tailed Student’s *t*-test, with significant values (*p* < 0.05) is displayed on the graph.

**Figure 3 biomedicines-11-03311-f003:**
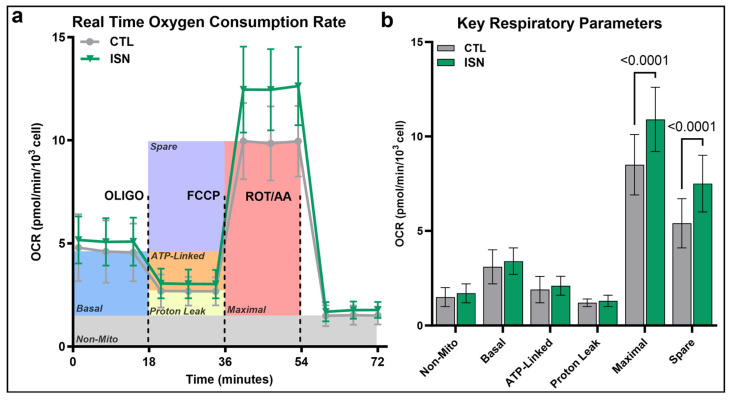
Real-time oxygen consumption rate (OCR) measurements during mitochondrial stress testing of primary osteoclast progenitors after 48 h of in vitro differentiation with 10 ng/mL of irisin treatment (ISN) or untreated controls (CTL) (**a**). Key respiratory parameters defined with respect to basal and non-mitochondrial respiration (**b**). Representative data are shown from one of three experimental repeats. *p*-value via 2-way ANOVA, with significant values (*p* < 0.05), is displayed on the graph.

## Data Availability

The data presented in this study and associated experimental repeats will be uploaded in the publicly available repository FigShare, DOI upon request to the corresponding author.

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
