# Peer review of "Irisin Enhances Mitochondrial Function in Osteoclast Progenitors during Differentiation"

_biomedicines, 2023, doi:10.3390/biomedicines11123311_

Round 1
Reviewer 1 Report
Comments and Suggestions for Authors
In this short report, Estell et al perform some basic evaluation of mitochondrial metabolism in osteoclast precursors treated with Irisin. They show that this myokine, which they had already shown increases osteoclast differentiation at the dose used here, rapidly increases mitochondrial metabolism, accompanied by an increase in mitochondrial content (mitotracker) and ROS (CellROX) at the 2d timepoint. The findings incrementally add to our understanding of the cell autonomous effects of Irisin on osteoclastogenesis. There are several things that are not adequately described.
1. The methods, text, and legend to not indicate the number of replicates for the imaging experiments. Were all replicates derived from cultures with the same osteoclast prep, or were they done from different animals and/or at different times?
2. The range of fluorescence intensities for mitotracker and ROS assays is very high. Since these cultures are heterogeneous and not all cells would go on to make mature osteoclasts, do the differences correlate with maturation? FACS analysis of these mononuclear cells is entirely feasible, and would not only show whether the whole population or just some cells shift, but could be paired with antibodies to indicate maturation such as beta3 integrin.
3. Was the Seahorse study only done once? Were the described 23 replicate wells all derived from the same initial cell population?
4. While the change in spare respiratory capacity did not reach statistical significance in the 1h group, the graph suggests that this is simply a statistical phenomenon, as the pattern is the same as for maximal OCR. Therefore, it is not appropriate to simply conclude that changes in spare capacity require 2 days.
5. The functional similarity (Seahorse study) of the effect of 1h vs 2d of irisin is surprising given the large effect of 2d on apparent mitochondrial content. This , ideally, should be explored further with mitotracker and cellROX at short (1-6 h timepoints), and mitochondrial content also assessed via other means such as expression of respiratory complex proteins and/or mitoDNA content.
Author Response
Thank you for your comments and fair evaluation of the impact of this work. While we agree the methods employed are basic and the contribution to the field only builds incrementally on our previous work, we felt this set of experiments was appropriate for publication as a Short Report when invited to preparate a manuscript for the upcoming special issue on myokines. Please find a point-by-point response to specific comments below, along with line-referenced changes in the revised manuscript.
- The methods, text, and legend do not indicate the number of replicates for the imaging experiments. Were all replicates derived from cultures with the same osteoclast prep, or were they done from different animals and/or at different times?
All primary cell experiments were conducted in duplicate using osteoclast preps from separate cell isolations at different times, with an additional experimental repeat conducted in the RAW cell line as detailed below. The data presented in the figures are shown as representatives from one of the experimental repeats, while the full set of data was originally intended to be uploaded separately as a supplement. An explanation of experimental repeats was unintentionally omitted from the methods section - these details have been clarified in the methods section (Lines 85-87) as well as the figure legends (Lines 151-152, 163-164, 187-188), and the full summary of data from all experiments have been organized into Supplemental Table 1 (Page 9) for ease of access.
- The range of fluorescence intensities for mitotracker and ROS assays is very high. Since these cultures are heterogeneous and not all cells would go on to make mature osteoclasts, do the differences correlate with maturation? FACS analysis of these mononuclear cells is entirely feasible, and would not only show whether the whole population or just some cells shift, but could be paired with antibodies to indicate maturation such as beta3 integrin.
As we have seen in our own work and in others in literature (eg. Reference 16) the variability for this type of measurement is quite high, due to the natural variation of mitochondrial content and the effect of cellular morphology concentrating the fluorescent dye in the area of smaller cells. To better account for the effect of area, we have reanalyzed the data to look at corrected total cell fluorescence (Lines 107-108) instead of simply mean fluorescence intensity for the region of interest drawn around a cell. This metric essentially normalizes fluorescence intensity by area, compensating for the effect of dye concentration in smaller cells. As you will note in Figures 1 and 2, the variability is still quite high for this metric, but the ability to analyze hundreds of cells across our technical replicates adequately powers the statistical analysis. The reproducibility of irisin’s effect across experimental repeats in two cell types further supports the original conclusions drawn.
We agree that some of the variability in fluorescence is likely due to the heterogeneity of the primary cell population and are currently pursuing the techniques mentioned (FACS paired with fluorescent antibodies for osteoclast markers) to look more specifically at osteoclast progenitors at different stages of maturation. This work is part of a broader effort to characterize shifts in metabolic program during osteoclastogenesis, and the mature osteoclast itself, and is still ongoing. In the present work, we chose a time point of 48 hours to capture the progenitor before any fusion has taken place, though maturation of individual progenitors towards that process will be an important nuance to address in future work. To address the limitation of heterogeneity in primary cells we have conducted additional experimental repeats in the RAW 264.7 cell line (Lines 88-90), which provides a homogenous macrophage population and has been used in the literature as osteoclast progenitors (eg. References 14, 15). Note that in Supplemental Figure 1 (Page 8), while the variability is still high, the significant effect of irisin is reproduced. We have also changed the format of the plots in Figures 1, 2, and S1 to show individual data points in order to better demonstrate the total number of cells analyzed.
- Was the Seahorse study only done once? Were the described 23 replicate wells all derived from the same initial cell population?
As mentioned in the prior comments, all primary cell experiments were conducted in duplicate from separate cell preparations, with an additional repeat in the RAW cell line to demonstrate reproducibility across cell type while addressing the aforementioned limitation of heterogeneity in primary cells. Within an experiment, the 23 technical replicate wells per group are all derived from the same initial cell population, and groups treated with the same media for 48 hours with or without irisin prior to imaging or metabolic analysis.
- While the change in spare respiratory capacity did not reach statistical significance in the 1h group, the graph suggests that this is simply a statistical phenomenon, as the pattern is the same as for maximal OCR. Therefore, it is not appropriate to simply conclude that changes in spare capacity require 2 days.
In the course of our additional experiments and revision we have chosen to omit the 1 hour irisin treatment group from the manuscript in order to focus the presentation of data to one treatment of irisin (continuous 48 hours). We agree the ability of 1 hour treatment to change additional respiratory parameters cannot be ruled out from the present data set, and are currently pursuing further experiments into the rapidity and timing of irisin signaling which this data will contribute to.
- The functional similarity (Seahorse study) of the effect of 1h vs 2d of irisin is surprising given the large effect of 2d on apparent mitochondrial content. Ideally, this should be explored further with Mitotracker and CellROX at short (1-6 h timepoints), and mitochondrial content also assessed via other means such as expression of respiratory complex proteins and/or mitoDNA content.
As mentioned above, we are currently pursuing these short-term treatments for a future publication, employing additional techniques such as downstream protein phosphorylation events, and we thank the reviewer for their suggestions on further experiments. We agree that additional means of quantifying mitochondrial content such as protein expression or mitoDNA content would bolster the present work, but feel confident the use of quantifiable, potentiometric dyes provides adequate evidence to complement the functional changes in respiration presented in this short report.
Reviewer 2 Report
Comments and Suggestions for Authors
The paper is interesting and well written, the key role of irisin on bone remodeling requires to be deepen by different authors. I have only some concerns that the authors fastly will solve:
- for each experiments in the legends should be detailed how many times has been performed
- I suggest a profound discussion about irisin and osteoclasts considering also other recent reports of the literature by other authors:
Int J Mol Sci 2023;24(12):9896. doi: 10.3390/ijms24129896. Int J Mol Sci 2021;22(19):10863. doi: 10.3390/ijms221910863.Author Response
Thank you for your kind comments on the paper, and for your suggestions to improve it. Please find a point-by-point response to specific comments below, along with line-referenced changes in the revised manuscript.
- For each experiment in the legends should be detailed how many times has been performed
An explanation of experimental repeats was unintentionally omitted from the methods section - these details have been clarified in the methods section (Lines 85-87) and added to the figure legends (Lines 151-152, 163-164, 187-188). A full summary of data from all experiments has been organized into Supplemental Table 1 (Page 9).
- I suggest a profound discussion about irisin and osteoclasts considering also other recent reports of the literature by other authors: Int J Mol Sci 2023;24(12):9896.doi: 10.3390/ijms24129896. Int J Mol Sci 2021;22(19):10863. doi: 10.3390/ijms221910863.
We have expanded the introduction and discussion to include these unintentionally overlooked works (Lines 262-269, References 22 and 23), thank you for your suggestions.
Reviewer 3 Report
Comments and Suggestions for Authors
The manuscript submmitted by Estell et al., is a nice piece of work describing how mitocondria is changed in osteoclast progrenitors due to the treatment with irisin, a well know myokine. A couple os isses need to be addressed:
1.- Do the authors represent mean+/-error or median? How many cells do authors take in consideration ofr average fluorescence? the error is quite big and it is impresive the grade of statistical signifficance they achieved. Please clarify how many cells per area do you sued for your calculations.
2.- In the discussion authors mention ´Similar experiments conducted in our lab featuring mature osteoclasts at later timepoints demonstrated comparatively low maximal respiration and spare capacity, with little effect of irisin treatment (not shown)´. I think this data should be shown at leas as supplemmetal day to get a better idea of the role of irisin in osteoclast differentiation.
3.- As several data indicate a role of irisin in both bone formation and resorption, d authors thin this miokine is a key regulator of bone remodeling?
Author Response
Thank you for your kind comments on the paper and for your suggestions to improve it. Please find a point-by-point response to specific comments below, along with line-referenced changes in the revised manuscript.
- Do the authors represent mean+/-error or median? How many cells do authors take in consideration for average fluorescence? the error is quite big and it is impressive the grade of statistical significance they achieved. Please clarify how many cells per area do you use for your calculations.
The data are quantified as the mean with standard deviation. Note that to better represent the number of cells analyzed in the imaging experiments, the format of plots in Figures 1 and 2 have been changed to show individual data points pooled from the replicate wells, with the mean represented as a black horizontal line. The format of bar graphs in Figure 3 remains unchanged – mean with bars showing standard deviation. For each imaging experiment, a fixed central region from 8 wells per group was analyzed with individual cells pooled across wells totaling between 500-2000 per group depending on cell density. These details have been added to the methods (Lines 105-106), and the specific numbers of cells analyzed for each group to the figure legends (Lines 150-151 and 163) and in Supplemental Table 1 (Page 9). As we have seen in our own work and in others in literature (eg. Reference 16) the variability for this type of measurement is quite high due to the natural variation of mitochondrial content and the effect of cellular morphology concentrating the fluorescent dye in the area of smaller cells. To better account for the effect of area, we have reanalyzed the data to look at corrected total cell fluorescence (Lines 107-108) instead of simply mean fluorescence intensity for the region of interest drawn around a cell. This metric essentially normalizes fluorescence intensity by area, compensating for the effect of dye concentration in smaller cells. As you will note in Figures 1 and 2, the variability is still quite high for this metric, but the ability to analyze hundreds of cells across our technical replicates adequately powers the statistical analysis. The reproducibility of irisin’s effect across experimental repeats in two cell types further supports the original conclusions drawn.
- In the discussion authors mention ´Similar experiments conducted in our lab featuring mature osteoclasts at later timepoints demonstrated comparatively low maximal respiration and spare capacity, with little effect of irisin treatment (not shown)´. I think this data should be shown at least as supplemental data to get a better idea of the role of irisin in osteoclast differentiation.
During further experiments and revision, we have decided not to include a discussion of this data in the manuscript. We are currently pursuing a broader effort to characterize shifts in metabolic program during osteoclastogenesis and the mature osteoclast itself, but this work is ongoing due to challenges in characterizing the inherently heterogenous population of mature osteoclasts and progenitors at later timepoints.
- As several data indicate a role of irisin in both bone formation and resorption, do authors think this myokine is a key regulator of bone remodeling?
In short yes, given the direct signaling effects of irisin demonstrated for both osteoclasts and osteoblasts, as well as on the osteocyte which regulates and coordinates their function. While the differing effects of irisin on bone formation and resorption that have been observed across a variety of experimental conditions may seem contradictory, we believe that this myokine plays a dynamic role in bone remodeling and that its net impact may change based on the timing and duration of signaling. We have expanded on the discussion to better address the relevance of the presented work to the larger understand of irisin’s role in bone by the field (Lines 262-269).
Round 2
Reviewer 1 Report
Comments and Suggestions for Authors
Thank you for all of your clarifications.
The first word on line 225 should be hours, not days.
Author Response
Thank you again for your thorough review of the manuscript, it certainly made a significant contribution to improving the final product. And thank you for catching this typo, it has been corrected (Line 225).